# Marketed Quinoa (*Chenopodium quinoa* Willd.) Seeds: A Mycotoxin-Free Matrix Contaminated by Mycotoxigenic Fungi

**DOI:** 10.3390/pathogens12030418

**Published:** 2023-03-06

**Authors:** Mara Quaglia, Giovanni Beccari, Giovanna Fabiana Vella, Riccardo Filippucci, Dario Buldini, Andrea Onofri, Michael Sulyok, Lorenzo Covarelli

**Affiliations:** 1Department of Agricultural, Food and Environmental Sciences, University of Perugia, Borgo XX Giugno, 74, 06121 Perugia, Italy; 2Department of Agrobiotechnology (IFA-Tulln), Institute of Bioanalytics and Agro-Metabolomics, University of Natural Resources and Life Sciences Vienna, Konrad Lorenz Strasse, 20, A-3430 Tulln, Austria

**Keywords:** mycobiota, mycotoxins, postharvest, quinoa

## Abstract

A total of 25 marketed quinoa seed samples different for origin, farming system and packaging were analyzed for the presence of mycotoxigenic fungi (by isolation both on Potato Dextrose Agar and with the deep-freezing blotter method) and relative contamination by mycotoxins (by LC-MS/MS analysis). Fungal microorganisms, but not mycotoxins, were detected in all the samples, and 25 isolates representative of the mycobiota were obtained. Morphological and molecular characterization and, for some isolates, the in vitro mycotoxigenic profile, allowed the identification of 19 fungal species within five different genera: *Alternaria*, *Aspergillus*, *Penicillium*, *Cladosporium* and *Fusarium.* Among the identified species, *Alternaria abundans*, *A. chartarum*, *A. arborescens*, *Cladosporium allicinum*, *C. parasubtilissimum*, *C. pseudocladosporioides*, *C. uwebraunianum*, *Aspergillus jensenii*, *A. tubingensis*, *Penicillium dipodomyis*, *P. verrucosum* and *P. citreosulfuratum* were first reported on quinoa, and *Alternaria infectoria* and *Fusarium oxysporum* were first reported on quinoa seeds. The geographical origin, farming system and packaging were showed to affect the amount and type of the isolated fungal species, highlighting that the level of fungal presence and their related secondary metabolites is conditioned by different steps of the quinoa supply chain. However, despite the presence of mycotoxigenic fungi, the marketed quinoa seeds analyzed resulted in being free from mycotoxins.

## 1. Introduction

Quinoa (*Chenopodium quinoa* Willd.) is an annual, dicotyledonous, seed-producing plant belonging to the *Amaranthaceae* family [1,2,3]. From the Andean region of Peru and Bolivia, considered its main center of origin [4,5,6], quinoa has spread to other parts of South, Central and North America and then to other continents, such as Europe, where it is cultivated in several countries, including France, England, Sweden, Denmark, Holland and Italy [4,5,7]. In particular, in Italy, quinoa was introduced and then validated as a good alternative to the traditional Mediterranean crop in the early 2000s [8,9,10,11].

This spread of quinoa was made possible thanks to its great adaptability to different cultivation environments. Indeed, quinoa can grow from 0 to 4.000 m above sea level, from the hot and dry climate of deserts to those cold and rainy of the high mountains and at variable photoperiods [4]. Obviously, the needs are different among the several genotypes. About 250 quinoa varieties are known [6], mainly distinct according to the pericarp or episperm colors (cream, yellow, orange, grey or cream/white, black and red), saponin contents (none, low, regular or high) and grain size (small, medium and large) [12].

The worldwide spread of quinoa cultivation has also been supported by the Food and Agriculture Organization of the United Nations (FAO), which firmly believes in the potential of this species to offer food security, in contrast to hunger and malnutrition; FAO declared 2013 as the “International Year of Quinoa” [7].

About the nutritional profile, quinoa is rich in carbohydrates, proteins and lipides, but also in vitamins, minerals (mainly potassium, magnesium and phosphorus) and other beneficial compounds, such as betaine, polyphenols, isoflavones and carotenoids [6]. Proteins contain all 10 essential amino acids and are rich in lysine and methionine, making quinoa the source of the best vegetal proteins, with a biological value similar to those of animal derivatives, such as beef and milk [6]. In addition, quinoa is gluten-free, hence suitable for celiac patients, and with a low glycemic index, that making it suitable for patients affected by type 2 diabetes [6].

Thanks to these beneficial properties for human consumption, not only has quinoa been chosen by the National Aeronautics and Space Administration (NASA) for the human diet during a space mission [6], but it has also been included in the list of healthy whole grains (WGs) by the HEALTH GRAIN Consortium [6,13]. As reported by Ross et al. [14], a WG is the kernel, intact or after grinding, cracking or flaking, from which the inedible parts (hull and husks) have been removed and in which the anatomical components (endosperm, germ and bran) are the same as in the intact grains or after they have undergone just a small loss (2% of the grain or 10% of the bran) during processing. To the traditional WGs (for example wheat, rice, barley, maize, rye, oats, millet, sorghum and triticale), also pseudocereals, such as amaranth, buckwheat, wild rice and quinoa, have been included over time [14]. A higher WG ratio in the diet is associated with lower risks of disease and mortality due to cardiovascular and neurological problems and cancer [14,15]. Despite this, the WG daily intake is still below the recommended dose in many countries, including Italy [14,15].

As a counterpart to the above-reported beneficial effects, a food safety threat can arise in quinoa from the presence of mycotoxigenic fungi and contamination by related mycotoxins. Mycotoxins are a structurally diverse group of mostly small-molecular-weight compounds able to cause a wide range of toxic effects in animals and humans. They are produced by the secondary metabolism of some filamentous fungi, which under suitable climatic conditions, may develop on the crop in pre-harvest and/or post-harvest [16,17].

The presence in quinoa seed of mycotoxigenic fungi mainly belonging to the genera *Aspergillus*, *Penicillium*, *Fusarium*, *Alternaria* and *Cladosporium* has already been reported by previous studies [18,19]. Contamination by aflatoxins (B_1_ and/or B_2_, G_1_ and G_2_) has been reported also in one baby food sample based on rice and quinoa [20] and in one quinoa flour sample [21]; moreover, multiple contaminations by fumonisins, trichothecenes and zearalenone have been reported in a mixed matrix of Andean cereal-like amaranth grains, including quinoa, kiwicha (*Amaranthus caudatus* L.) and kañiwa (*Chenopodium pallidicaule* Aellen) [22]. However, being just an ingredient in these matrices, the impact of quinoa on the contamination is uncertain. In unprocessed quinoa seeds, despite the presence of mycotoxigenic fungi, no mycotoxins were detected by Pappier et al. [18], while Ramos-Diaz et al. [23] detected the presence of mycotoxins, mainly from the *Alternaria*, *Cladosporium*, *Fusarium* and *Penicillium* species. However, Ramos-Diaz et al. [23] also showed that cleaning procedures applied for saponin removal significantly reduced mycotoxin levels, bringing them to levels below the detection limits.

Considering this, a survey on marketed quinoa seeds was realized to monitor the presence of mycotoxigenic fungi, as well as possible mycotoxin contaminations, to assess if ready-to-use quinoa seeds are a mycotoxin-free matrix, and therefore, the beneficial properties of this food are not counteracted by this dangerous threat for consumers’ health.

## 2. Materials and Methods

### 2.1. Marketed Quinoa Seed Samples

A total of 25 marketed quinoa seed samples were analyzed in this study. Among these, 20 samples, of 300 g each, were purchased between September 2017 and February 2018 in grocery stores in the Perugia area (Umbria, Central Italy) and chosen as representative of all types of marketed quinoa available in this area. Another five samples, of the same weight, were purchased in the same period in South American markets. Samples were stored at 4 °C before the analysis. The 25 samples differed in geographical origin, seed color, farming system and packaging (Table 1).

In detail, 20 samples, equal to 80% of the total, were of extra EU origin, and of these, 9 came from Peru (1, 4, 6, 11, 13–16 and 20), 4 from Bolivia (18, 19, 21 and 25), 2 from Ecuador (23 and 24) and 1 from Chile (22), while of the remaining 4 (5, 9, 10 and 12), the country of origin was not specified. The other 5 samples, equal to 20% of the total, were of EU origin, and of these, 4 samples (2, 3, 7 and 8) came from Italy, while of the remaining ones (17), the country of origin was not specified. Considering their color, 22 seed samples (1–13, 16, 17 and 19–25) were white, 2 black (14 and 18) and only 1 was red (15). Finally, 13 samples (1–3, 5, 7, 8, 10, 11, 13–16 and 21) came from organic farming, whereas the other 12 (4, 6, 9, 12, 17–20 and 22–25) from an Integrated Pest Management (IPM) system. Except for sample 10, packaged in unsealed cardboard boxes, 18 samples (1–9 and 12–20) were packaged in sealed plastic bags, while 6 of them (11 and 21–25) were marketed in bulk.

### 2.2. Isolation and Molecular Identification of the Fungal Microorganisms Associated with the Marketed Quinoa Seed Samples

For each of the 25 quinoa samples, 200 seeds (about 0.4 g) were randomly chosen. Of these, 100 seeds were placed into Petri dishes (9 cm diameter) on two layers of sterilized filter paper soaked in 5 mL of sterilized deionized water and used for the deep-freezing blotter (DFB) test, following the protocol of Limonard [24], while the other 100 were surface disinfected and placed into Petri dishes (9 cm diameter) containing Potato Dextrose Agar (PDA, Biolife Italiana, Milan, Italy), (pH 5.7), as described by Covarelli et al. [25]. According to Beccari et al. [26], for each sample, 10 plates (replicates) per method (DFB test or isolation on PDA) each containing 10 seeds were used, for a total of 100 seeds per method per sample. All plates were kept at 21 ± 2 °C, in the dark. To morphologically identify the fungal genera associated with each sample, visual observations were carried out on each plate after 7 days of incubation with a stereomicroscope (SZX9, Olympus, Tokyo, Japan) and a light microscope (Axiophot Zeiss, Carl Zeiss, Jena, Germany), in the latter case after mounting slides in sterile deionized water.

According to Quaglia et al. [27], each colony chosen as representative of a well-defined fungal morphotype, based on color, morphology, growth and microscopic features, was transferred onto new PDA plates and incubated at 21 ± 2 °C, in the dark for 7 days. This selection allowed obtaining, from the total of the colonies developed on the 25 samples, a subset of 25 representative isolates, each corresponding to one morphotype.

Monosporic cultures of each of the 25 representative isolates were obtained and successively cultured in the dark at 21 ± 2 °C, on an orbital shaker at 130 rpm, in the Czapek Yeast Broth (CYB) medium [27,28,29]. After 14 days of incubation, each fungal culture was filtered through sterile filter paper to obtain mycelia separation from the liquid media. After collection, mycelia were transferred into 0.5 mL plastic tubes (Eppendorf, Hamburg, Germany) and further used for DNA extraction, performed from 0.01 g of freeze-dried mycelia, as described by Covarelli et al. [25]. The representative isolates were also kept in the mycological collection of the Department of Agricultural, Food and Environmental Sciences of the University of Perugia (Perugia, Italy) and stored at −80 °C.

To molecularly identify the 25 representative isolates obtained in this study, a phylogenetic analysis was performed. In detail (Appendix A):for those isolates morphologically ascribed to genera *Alternaria* and *Penicillium*, the molecular characterization was carried out using the sequences of the *Internal Transcribed Spacer* (*ITS*) region combined with those of the *RNA polymerase II largest subunit* (*RPB2*) region [30,31,32];for those isolates morphologically ascribed to the genus *Cladosporium*, the sequences of the *ITS* region combined with those of the *actin* (*ACT*) gene [33] were used;for those isolates morphologically identified as belonging to the genus *Aspergillus*, the combined sequences of the *β-tubulin* (*BenA*) and *calmodulin* (*CaM*) genes [34,35] were adopted;finally, for those isolates which belonged to the genus *Fusarium*, the *Translation Elongation Factor 1*-α sequence (*TEF1-α*) was used [36,37,38,39].

The primers adopted in the PCR assays are listed in Appendix A [36,38,39,40,41,42,43,44].

As reported by Quaglia et al. [27], PCR reactions were carried out using a T100^TM^ Thermal Cycler (Biorad, Foster City, CA, USA) in a total volume of 50 μL, 5 μL of which was of the DNA working solution containing 15 ng μL^−1^ of DNA, for a total of 75 ng of template DNA for each reaction, and following the cycling profiles described in Appendix A [36,41,43,45]. Amplification products were separated at 110 Volt for 30 min in a 2% agarose gel in TAE buffer 1X with RedSafe^TM^ (4% *v*/*v*) (Chembio, Medford, NY, USA) and sent to Genewiz Europe (Azenta Life Sciences, Leipzig, Germany) for purification and sequencing. Each consensus sequence was preliminarily examined by MEGA software version 7.0 [46], using the View Sequencer File (Trace Editor) functionality. Successively, the sequences were subject to *Basic Local Alignment Search Tool* (BLAST) [47] analysis and finally deposited in GenBank (Appendix A).

MEGA software 7.0 was also used to perform phylogenetic analyses, which were based on a single dataset of *TEF1-α* sequences for *Fusarium* spp. and on a combined dataset of *ITS* and *RPB2* (*Alternaria* spp. and *Penicillium* spp.), *ITS* and *ACT* (*Cladosporium* spp.) and *BenA* and *CaM* (*Aspergillus* spp.). For each genus, the sequences of the isolates obtained in the present investigation, as well as those of representative isolates obtained from Genbank, including those of the main *Alternaria* (Appendix A), *Penicillium* (Appendix A), *Aspergillus* (Appendix A), *Cladosporium* (Appendix A) and *Fusarium* (Appendix A) species reported in the literature on *C. quinoa* (Appendix A) [18,19,48,49,50,51,52,53,54,55], were used. Moreover, *Stemphylium herbarum* CBS 191.86 [31], *Talaromyces flavus* CBS 310.38 [30], *Cercospora beticola* CBS 116,456 [56] and *Neurospora crassa* OR74A [57] were included as outgroups for phylogenetic analysis of *Alternaria*, *Aspergillus* and *Penicillium*, *Cladosporium* and *Fusarium*, respectively. Single (*Fusarium*) or concatenated (other genera) sequences were aligned, nucleotide gaps and missing data were deleted, and phylogenetic trees were built using the Neighbor-Joining method [58], with the bootstrap test for 1000 replicates [59]. The evolutionary distances were computed using the Maximum Composite Likelihood method [60].

### 2.3. Fungal Secondary Metabolites Analysis in the Marketed Quinoa Seed Samples and in Fungal Cultures

Analysis of fungal secondary metabolites was performed on each of the 25 marketed quinoa seed samples and, additionally, also on fungal cultures of isolates Q 145 for which the phylogenetic analysis had not led to species identification, and Q 146, to assess its ability to produce aflatoxins. For detection of secondary metabolites on fungal culture filtrates, each fungal isolate was grown for 10 days at 23 ± 2 °C, in the dark, on Petri dishes (9 cm diameter) containing 20 mL of Czapek Yeast Autolysate (CYA) Agar medium (HiMedia Laboratories, GmbH, Einhausen, Germany) [61], stored at −80 °C for 2 h, freeze-dried for 24 h with a Heto Power Dry LL3000 (Thermo Fisher Scientific, Waltham, Ma, USA) and finely ground with a mixer mill (MM200, Retsch, Verder Scientific, Haan, Germany).

Detection of fungal secondary metabolites was performed on 5 g of each quinoa seed subsample and 5 g of each freeze-dried fungal culture by liquid chromatography-tandem mass spectrometry (LC-MS/MS), using the method described by Sulyok et al. [62].

### 2.4. Statistical Analysis

For each plate, the number of seeds from which each morphotype had developed was analyzed using a generalized linear model with Poisson error and log-link. Species and the detection method (together with their interactions) were included as the predictors, and the back-transformed means, together with ‘delta’ standard errors [63], were derived and compared by using a multiple comparison testing procedure with multiplicity adjustment, as suggested by Bretz et al. [64].

For the incidence of each fungal genera and total fungi, the effects of packaging (bulk, unsealed cardboard box and sealed plastic box) and farming system (IPM and organic farming) were used as the predictors in a generalized linear model with Poisson error and log-link. The significance of each effect was evaluated using likelihood ratio tests.

Statistical analyses were performed in the R statistical environment [65], together with the packages multcomp [66] and emmeans [67].

## 3. Results

### 3.1. Mycobiota Associated with Marketed Quinoa Seed Samples

Considering the combination of the 2 isolation methods, after 1 week of incubation, fungal development was detected in all (100%) quinoa samples, with a total of 418 colonies obtained. Fungal microorganisms of the genera *Alternaria*, *Aspergillus*, *Cladosporium*, *Fusarium* and *Penicillium* were morphologically identified, with the highest incidence recorded for the *Alternaria* genus (130 colonies, corresponding to 31.1% of the total), followed by *Cladosporium* (86 colonies, 20.6%), *Penicillum* (69 colonies, 16.5%), *Fusarium* (62 colonies, 14.8%) and *Aspergillus* (27 colonies, 6.5%). The remaining 44 colonies (10.5% of the total) belonged to “other” fungal genera and were mainly represented by *Epicoccum* spp. (14 colonies), *Botrytis* spp. (10 colonies), *Trichoderma* spp. (5 colonies), *Rhizopus* spp. (3 colonies) and *Talaromyces* spp. (1 colony) (Figure 1).

*Alternaria* was detected in 19 (1–3, 6–8, 10, 12–20 and 23–25) of the 25 quinoa samples (Table 2). In 17 of the 19 samples (except samples 16 and 24) positive to *Alternaria*, the co-occurrence of *Cladosporium* (Table 2), which was additionally present in another 5 samples (4, 5, 11, 21 and 22), was also recorded. *Penicillium* was detected in 14 samples (1, 2, 4, 6, 8–10, 13–16, 20, 21 and 23), while *Aspergillus* was detected in 11 samples (5–8, 10, 11, 15, 16, 20, 23 and 24). Co-occurrence of these 2 latter genera was detected in 7 samples (6, 8, 10, 15, 16, 20 and 23) (Table 2). *Fusarium* was detected only in 4 samples (5, 6, 15 and 20). In 3 analyzed samples (6, 15 and 20), all 5 genera were simultaneously present, whereas in the other 3 samples (8, 10 and 23), the co-occurrence of 4 genera (*Alternaria*, *Cladosporium*, *Aspergillus* and *Penicillium*) was observed (Table 2).

Considering the globality of the samples analyzed, significant differences between the two isolation methods adopted in this study for the average number of the total fungi developed from marketed quinoa seeds were recorded. In detail, the DFB test allowed us to obtain a significantly (*p* < 0.05) greater average number of total isolated fungi (1.23 ± 0.07) compared to those obtained by isolation on PDA (0.44 ± 0.04).

Additionally, significant differences were observed also within each isolation method among fungal genera and within each fungal genus between the two isolation methods adopted (Figure 2). For example, the DFB method allowed the obtainment of a significantly higher number of *Alternaria*, *Penicillium*, *Fusarium* and “other” fungal colonies than the PDA method. On the contrary, isolation on PDA allowed isolation of a significantly higher number of *Aspergillus* colonies. Finally, no significant difference occurred between the two isolation methods in the average numbers of *Cladosporium* colonies.

Analyzing the marketed quinoa samples individually, significant differences emerged in the average number of colonies of the total fungi or each fungal genus obtained by isolation on PDA, by the DFB method or by the two isolation methods taken together (total) (Appendix A). For example, samples 4, 5, 6, 10 and 17 showed the significantly highest incidence of total fungi; samples 5 and 6 showed the significantly highest incidence of *Alternaria* spp. and *Fusarium* spp.; sample 4 showed the significantly highest incidence of *Penicillium* spp.; and samples 17 and 20 showed the significantly highest incidence of *Cladosporium* spp.

### 3.2. Phylogenetic Analysis of Isolates Developed from Marketed Quinoa Seeds

Following the criteria explained in Section 2.2, a total of 25 isolates were collected as representative of all the observed morphotypes. Of these 25 isolates, 7 isolates (Q 54, Q 113, Q 132, Q 149, Q 178, Q 180 and Q 184) were morphologically identified as *Alternaria* spp., 6 isolates (Q 5, Q 9, Q 35, Q 39, Q 145 and Q 181) as *Penicillium* spp., 7 isolates (Q 55, Q 61, Q 77, Q 92, Q 111, Q 131 and Q 162) as *Cladosporium* spp., 4 isolates (Q 29, Q 49, Q 73, and Q 146) as *Aspergillus* spp. and 1 isolate (Q 185) as *Fusarium* spp. For each isolate, BLAST analysis of the above-reported amplified regions (Section 2.2) confirmed the morphological identification (Appendix A). Moreover, phylogenetic analysis led to the definition of the species as follows.

In the phylogram constructed on the concatenated sequences of the *ITS* and *RPB2* regions of the *Alternaria* species, according to Woudenberg et al. [31], four major clades emerged (Figure 3): clade A (which included species of the section *Alternata*); clade B (section *Pseudoulocladium*); clade C (section *Chalastospora*); and clade D (section *Infectoriae*). Isolates Q 54, Q 113 and Q 184 clustered in clade A; isolate Q 54 clustered together with strain CBS 102,605 of *A. arborescens*; while isolates Q 113 and Q 184 clustered together with isolate CBS 916.96 of *A. alternata*. Isolates Q 178 and Q 180 clustered in clade B together with strain CBS 200.67 of *A. chartarum*. Isolate Q 132 clustered in clade C together with isolate CBS 534.83 of *A. abundas*, and isolate Q 149 clustered in clade D together with isolate CBS 210.86 of *A. infectoria*.

*Alternaria alternata* and *A. arborescens* were the most prevalent species, both detected by the two methods adopted in this study (Figure 4A). *A. alternata* (63 total colonies) was detected in 7 samples (6, 8, 17–20 and 25) by the DFB method (57 colonies), and in 3 samples also by PDA (6 colonies; samples 20 and 25). *A. arborescens* (55 total colonies) was detected in 11 samples (1–3, 7, 8, 10 and 12–16) by the DFB method (51 colonies), and in 3 of them (3, 13 and 16), also by PDA (4 colonies). *Alternaria chartarum* (7 total colonies, 6 of which were obtained by the DFB method from samples 23 and 24, and 1 by PDA from sample 25), *A. infectoria* (4 total colonies, all obtained from sample 4 by PDA) and *A. abundans* (1 colony obtained by the DFB method on sample 17) were the least represented species (Figure 4A).

Using the concatenated sequences of the *ITS* and *RPB2* regions, according to Houbraken and Samson [30] and Visagie et al. [68], seven major clades emerged in the phylogram of the *Penicillium* species (Figure 5): clade A, which included species of the section *Fasciculata*; clade B (section *Penicillium*); clade C (section *Chrysogena*); clade D (section *Canescentia*); clade E (section *Brevicompacta*); clade F (section *Exilicaulis*); and clade G (section *Citrina*).

Considering the isolates obtained in this study, isolates Q 5 and Q 145 clustered in clade A (Figure 5). However, while Q 5 clustered together with isolate CBS 603.74 of *P. verrucosum*, Q 145 clustered together with isolates CBS 390.48 of *P. viridicatum*, CBS 101486 of *P. freii*, CBS 324.89 of *P. aurantiogriseum*, F 727 of *P. cellarum* and CBS 222.28 of *P. polonicum*. Therefore, to refine the identification of isolate Q 145, LC-MS/MS analysis was performed on its freeze-dried culture. The detection, among the other fungal secondary metabolites, of the viridicatols, such as cyclopenin, cyclopenol, cyclopeptine, dehydrocyclopeptine and o-methylviridicatin, and of the two diketopiperazines puberulin A and rugulusuvin (Table 3), led to excluding the species *P. viridicatum*, *P. aurantiogriseum* and *P. cellarum* as unable to biosynthesize these compounds [69,70]. Between the species *P. freii* and *P. polonicum*, both able to biosynthesize the above reported compounds [69,70], in the *BLAST* analysis isolate, Q 145 show the greatest sequences similarity to the *P. polonicum* species (Appendix A). Isolates Q 9, Q 39 and Q 181 clustered in clade C (Figure 5). In detail, isolates Q 9 and Q 181 clustered together with isolate CBS 306.48 of *P. chrysogenum* and isolate Q 39 with isolate CBS 110,412 of *P. dipodomyis*. Isolate Q 35 clustered in clade F, together with isolate PUMCH_Q141 of *P. toxicarium* and isolates CBS 139,158 and CBS 139,162 of *P*. *citreosulfuratum* (Figure 5) As reported by Visagie et al. [71], *P. toxicarium* and *P. citreosulfuratum* are the same species, for which *P. citreosulfuratum* represents the correct name. For this reason, our isolate Q 35 is here named *P. citreosulfuratum* (Figure 5).

Within the *Penicillium* genus, *P. polonicum*, with 34 total colonies, was the most abundant species (Figure 4C). It was obtained from 7 samples (4, 6, 8–10, 13 and 15): from 5 samples with the use of the PDA method (13 colonies) and from the other 2 samples with the use of the DFB method (21 colonies). This species was followed by *P. dipodomyis* (18 total colonies, 9 of which were obtained with PDA and the other 9 with the DFB method), *P. chrysogenum* (10 total colonies, 7 of which were obtained with the DFB method and the other 3 isolated on PDA), *P. verrucosum* (6 total colonies, all obtained with the DFB method) and *P. citreosulfuratum* (1 colony obtained by isolation on PDA).

Using the concatenated sequences of the *CaM* and *BenA* regions, according to Varga et al. [34] and Houbraken et al. [72], six major clades emerged in the phylogram of the *Aspergillus* species (Figure 6): clade A, which included species of the section *Flavi*; clade B (sections *Circumdati*/*Terrei*); clade C (section *Fumigati*); clade D (sections *Usti*/*Nidulantes*); clade E (section *Nigri*, with the two subclades *tubingensis* and *awamori*/*welwitschiae*); and clade F (section *Flavipedes*).

Isolate Q 29 clustered in clade D together with isolates NRRL 58600, UTHSC 09_425 and NRRL 225 of *A. jensenii*; isolate Q 49 clustered in clade C with isolate CBS 133.61 of *A. fumigatus*; isolate Q 73 clustered in clade E, subclade *tubingensis*, together with isolate CBS 134.48 of *A. tubingensis*; and isolate Q 146 clustered in clade A, together with isolates CBS 100.927 of *A. flavus*. The isolate Q 146 was also capable of producing aflatoxins B_1_, B_2_, M_1_ and P_1_ and their precursors sterigmatocystin and O-methylsterigmatocystin (Appendix A). Moreover, for isolates Q 49 and Q 146, A. *fumigatus* and A. *flavus* were the only species resulting, respectively, from the *BLAST* analysis (Appendix A). Within the genus *Aspergilllus* (Figure 4E), *A. flavus* (13 total colonies, 12 of which were obtained by isolation on PDA from samples 5, 7, 8, 10 and 20, and 1 from sample 16 with the DFB method) and *A. tubingensis* (8 total colonies, 7 of which were obtained with the DFB method from samples 6, 10, 11, 15, 23 and 24, and 1 on PDA from sample 8) were the most abundant species. Instead, *A. fumigatus* (5 colonies, all isolated on PDA from sample 16) and *A. jensenii* (2 colonies isolated on PDA from sample 10) were the least represented species.

Using the concatenated sequences of the *ITS* and *ACT* regions, according to Bensch et al. [33,56], two major clades emerged in the phylogram of the *Cladosporium* species (Figure 7): clade A, which included species of the *C. cladosporioides* complex, and clade B, which included species of the *C. herbarum* complex. Five isolates obtained in this survey clustered in clade A and two in clade B. Inside clade A, isolates Q 55 and Q 162 clustered together with isolate CBS 112,388 of *C. cladosporioides*, isolates Q 92 and Q 131 clustered together with isolate CBS 125,993 of *C. pseudocladosporioides* and, finally, isolate Q 111 clustered with isolate CBS 139,572 of *C. uwebraunianum*. Inside clade B, isolate Q 61 clustered with isolate CBS 121,624 of *C. allicinum* and isolate Q 77 clustered with isolate CBS 143,361 of *C. parasubtilissimum*.

Within the genus *Cladosporium* (Figure 4B), *C. cladosporioides* (23 total colonies, 7 of which were obtained with the DFB method from samples 6 and 8, and 16 colonies isolated on PDA from samples 5–8, 2, 22, 23 and 25) and *C. parasubtilissimum* (23 total colonies, 12 of which were obtained with the DFB method from samples 11, 12 and 15, and 11 colonies on PDA from samples 17 and 18) were the prevalent species. Lower incidence was recorded for the species *C. allicinum* (16 total colonies, 15 of which were obtained with the DFB method from samples 1–3, 7 and 10, and 1 colony isolated on PDA from sample 8) and *C. uwebraunianum* (15 total colonies, 11 of which were isolated on PDA from sample 20, and 4 colonies obtained with the DFB method from samples 13, 18 and 20).

Using the sequence *TEF1-α* region, four major clades (A-D) emerged in the phylogram of the *Fusarium* species (Figure 8). According to Lombard et al. [73] and Crous et al. [74], clade A contains species of the *Fusarium oxysporum* and *Fusarium fujikuroi* species complex (FOSC and FFSC, respectively). According to Laraba et al. [75], clade B contains species of the *Fusarium tricinctum* species complex (FTSC) and clade C of the *Fusarium sambucinum* species complex (FSAMSC). Lastly, according to Castellá and Cabañes [76], clade D was identifiable as the *Fusarium incarnatum-equiseti* species complex (FIESC). Our isolate Q 185 clustered in clade A, together with isolate CBS 144,134 of *F. oxysporum*. This species was recorded mainly following DFB (60 colonies from samples 5, 6 and 15) and with a very low incidence on PDA (2 colonies from sample 20).

As previously mentioned, the samples analyzed in this study differed in geographical origin, coming partly from Extra-European (Extra EU) countries (including those of Central and South America, such as Bolivia, Chile, Ecuador and Peru) and partly from European (EU) countries, including Italy (Table 1). The average number of colonies of total fungi significantly varied with the geographical origin of the samples, with the highest incidence recorded in samples from both EU and Extra-EU origin, including those from Peru, and the lowest incidence detected in samples from Chile (Figure 9A). In particular, samples from the unspecified EU country showed the highest average number of *Alternaria* and *Cladosporium* colonies, samples from Extra UE countries showed the highest *Fusarium* incidence, and the samples from Peru had the highest incidence of *Penicillium* and *Fusarium* (Figure 9A).

Also, the farming system showed a significant effect on the incidence of isolated fungi. In detail, a significantly higher number of colonies of total fungi was recorded in the organic samples with respect to IPM samples (Figure 9B). Considering the individual fungal genera, in the organic farming system, the isolation incidence was significantly higher for the fungi of the genera *Cladosporium*, *Penicillium* and “other”. On the contrary, the incidence of *Aspergillus* colonies was significantly higher in the IPM system (Figure 9B).

Regarding packaging, a significantly higher number of colonies of total fungi developed from quinoa seeds marketed in an unsealed cardboard box (Figure 9C). This primacy was mainly justified by the greater incidence, in this sample, of fungi of the genus *Alternaria*. Surprisingly, the seed samples marketed in bulk were those presenting the lowest fungal contaminations (Figure 9C).

### 3.3. Secondary Metabolites in the Marketed Quinoa Seed Samples

According to Sulyok et al. [62], LC-MS/MS is a method that allows simultaneous detection and quantification in a food matrix of more than 500 fungal secondary metabolites, including main mycotoxins (i.e., aflatoxins, ochratoxins and deoxynivalenol), whose maximum presence in food is limited by the law, as well as emerging mycotoxins and other compounds. The analysis did not show the presence of any mycotoxins regulated by current EU legislation nor any emerging mycotoxins in the 25 marketed quinoa seed samples investigated in this study. However, samples 21–25 exhibited low levels (1–31 µg kg^−1^) of the *Alternaria* metabolite altersetin, accompanied by diketopiperazines and tryptophol at similar levels (Table 4).

## 4. Discussion

Andean grains, including quinoa, are excellent food alternatives to cereal grains in some food preparations, and for this reason, they can also be called pseudocereals.

Despite the abundant literature concerning the infection of cereals by mycotoxigenic fungi and relative contamination by mycotoxins, the number of scientific papers concerning the presence of these contaminants in pseudocereals, quinoa in particular, is smaller [18,19,20,21,22,48,49,50,51].

For this reason, in the present survey, the presence of mycotoxigenic fungi and mycotoxins was analyzed on 25 quinoa seed samples purchased between September 2017 and February 2018 in grocery stores of the Perugia area (Umbria, Central Italy) and regarded as representative of all types of marketed quinoa available in this area. In addition, five samples were purchased from South American markets (Bolivia, Chile and Ecuador) and included in this investigation.

In this study, as also shown by previous researchers [18,77,78], fungal contamination was detected in all the analyzed samples, with the genera *Alternaria*, *Cladosporium*, *Penicillium*, *Fusarium* and *Aspergillus* as the main components of the quinoa seed mycobiome. In particular, *Alternaria* was the most detected genus, as previously found in quinoa seed samples harvested in Brazil [78], the Czech Republic and Peru [77]. The genus *Aspergillus*, prevalent in samples harvested in Argentina [18], was the least frequent in the samples analyzed in this study. This difference could be explained considering that, usually, *Aspergillus* species are external contaminants of quinoa seeds, which could be removed during the technological processes, such as the saponin removing procedure, to which quinoa seeds are subject before commercialization. On this basis, it is possible to hypothesize that saponin removal causes a proportional increase of the fungal species associated with the internal mycobioma of the analyzed matrix [18,23]. About *Alternaria*, a close association with the genus *Cladosporium* was found, since *Cladosporium* was detected in about 90% of the samples contaminated by *Alternaria*. A similar association was found by Sacco et al. [21] for *Aspergillus* and *Penicillium* in cereal and pseudocereal flours. According to these authors, the association of different fungal genera excludes negative competition between them. The *Alternaria*–*Cladosporium* association is also reported in the literature as responsible for plant diseases [79]. For this reason, it would be interesting to investigate in-depth the interactions between these two fungal genera to understand what is their mutual advantage and which biochemical and molecular factors are involved, also in relation to the substrate.

The molecular approach allowed not only to confirm genus identification based on morphological features but also to identify the species to which representative morphotypes belong. Thus, inside the genera *Alternaria*, *Cladosporium*, *Aspergillus* and *Fusarium*, the species *A. abundans* (isolate Q 132), *A. alternata* (isolate Q 113 and Q 184), *A. chartarum* (isolates Q 178 and Q 180), *A. infectoria* (isolate Q 149), *A. arborescens* (isolate Q 54), *A. flavus* (isolate Q 146), *A. fumigatus* (isolate Q 49), *A. jensenii* (isolate Q 54), *A. tubingensis* (isolate Q 73), *C. allicinum* (isolate Q 61), *C. cladosporioides* (isolates Q 55 and Q 162), *C. parasubtilissimum* (isolate Q 77), *C. pseudocladosporioides* (isolates Q 92 and Q 131), *C. uwebraunianum* (isolate Q 111) and *F. oxysporum* (isolate Q 185), were identified. The attribution of isolate Q 146 to the species *A. flavus* is of fundamental importance, given that it is a notoriously aflatoxigenic species, as also showed by the biochemical characterization of this isolate. For the identification of *Penicillium* isolates, the molecular approach was also coupled, when necessary, with biochemical characterization and allowed the identification of *P. citreosulfuratum* (isolate Q 35), *P. chrysogenum* (isolates Q 9 and Q 181), *P. dipodomyis* (isolate Q 39), *P. polonicum* (isolate Q 145) and *P. verrucosum* (isolate Q 5).

As previously reported by Kozlovskii et al. [69] and Quaglia et al. [27], chemotaxonomy here confirmed its usefulness as support in the characterization of *Penicillium* species, when morphological and/or molecular features did not lead to univocal results. Indeed, in the phylogenetic tree constructed with the concatenated sequences of the *ITS* and *RPB2* regions, isolate Q 145 clustered inside the *Fasciculata* clade with isolates of the species *P. aurantiogriseum*, *P. cellarum*, *P. freii*, *P. polonicum* and *P. viridicataum*. The detection in the freeze-dried culture filtrate, by LC-MS/MS analyses, of the viridicatols (cyclopenin, cyclopenol, cyclopeptine, dehydrocyclopeptine and O-methylviridicatin) and of the two diketopiperazines (puberulin A and rugulusuvin) allowed excluding the species *P. aurantiogriseum*, *P. cellarum* and *P. viridicataum* as unable to produce them [69,70] and to narrow the choice to the species *P. freii* and *P. polonicum*. The higher sequences’ similarities with *P. polonicum* allow us to identify isolate Q 145 as belonging to this species.

While *A. alternata*, *A. infectoria*, *C. cladosporioides*, *F. oxypsorum*, *A. flavus*, *A. fumigatus*, *P. chrysogenum* and *P. polonicum* have been previously reported on *Chenopodium quinoa* [18,48,50], to the best of our knowledge, this is the first world report of *A. abundans*, *A. chartarum*, *A. arborescens*, *C. allicinum*, *C. parasubtilissimum*, *C. pseudocladosporioides*, *C. uwebraunianum*, *A. jensenii*, *A. tubingensis*, *P. citreosulfuratum*, *P. dipodomyis* and *P. verrucosum* on this crop. In addition, the above-reported *A. infectoria* and *F. oxysporum* were previously detected in plants/roots and not in seeds, as detected in this survey [48,50].

The geographical origin of the samples showed a significant effect on the amount of the total isolated fungi. However, the absence of significant differences was detected on the base of the EU and extra-EU origin. On the contrary, differences between EU or extra-EU origin were recorded in qualitative terms, with the maximum biological variability associated with the quinoa sample of extra-EU origin, from which 18 of the 19 fungal species identified in the present work had been isolated, with the only exception *A. abundans*. A lower biological variability was associated with the quinoa samples of the EU origin, where 11 fungal species were obtained (*A. alternata*, *A. arborescens*, *A. abundans*, *P. polonicum*, *P. dipodomyis*, *A. flavus*, *A. tubingensis*, *C. cladosporioides*, *C. parasubtilissimum*, *C. allicinum* and *C. uwebrauinianum*) and the *Fusarium* species were not detected. In regard to *Alternaria* spp., although the highest number of colonies were detected in samples of extra-EU origin, the highest number of species were recorded in EU samples. About *Penicillium*, extra-EU quinoa samples showed both the highest number of colonies and species. Thus, in quantitative terms, it is not possible to discriminate between marketed EU and extra-EU samples for the infection level, but it is clear that a double number of fungal species were recorded in extra-EU samples. To our knowledge, there is no available literature in which the presence of mycotoxigenic fungi in EU and extra-EU quinoa is compared. Instead, a comparative study on mycotoxin occurrence in EU and extra-EU quinoa samples was recently published by Ramos-Diaz et al. [23]. In this study, the authors showed lower mycotoxin contamination in Andean grains (quinoa and kañiwa) cultivated around the center of the origin of the species than those cultivated in Northern Europe. However, looking at the data in detail, quinoa samples from the EU (Denmark) differed from quinoa samples from extra-EU origin (Bolivia and Peru) only for the presence at levels > 1000 μg kg^−1^ of the *Fusarium* secondary metabolites antibiotic-Y and aurofusarin and of the *Alternaria* mycotoxin tenuazonic acid. Indeed, a comparable level of contamination (100–1000 μg kg^−1^) with the *Fusarium* secondary metabolites butenolide, culmorin and equisetin; the *Alternaria* mycotoxin altersetin; and the *Cladosporium* secondary metabolites calphostin and emodin was found in quinoa samples, both of EU (Danmark) and extra-EU (Bolivia and Peru) origin. Finally, low contamination levels (<99 μg kg^−1^) of ochratoxins A (OTA) and B, atlantinon A and questomycin A (typically produced by *Penicillium*), and cladosporin (typically produced by *Cladosporium*) were detected only in quinoa samples of extra-EU origin (Bolivia and Peru). Although based on two different subjects (mycotoxigenic fungi and mycotoxins), from the comparison of the results of the present survey with those of Ramos-Diaz et al. [23], the previous statement that EU and extra-EU quinoa samples appear not to have particular differences in mycotoxin contamination is reinforced. Moreover, even if *Fusarium* spp. were not detected in the EU samples analyzed in this investigation, the presence of *Fusarium* secondary metabolites reported by Ramos-Diaz et al. [23] in quinoa seeds from Denmark showed that also *Fusarium* species can be associated with EU quinoa seed samples, as also previously reported by Beccari et al. [51]. However, both Ramos-Diaz et al. [23] and Beccari et al. [51] refer to the species *Fusarium equiseti*, while in this study, the non-mycotoxigenic species *Fusarium oxysporum* was identified in four extra-EU samples.

The current European legislation has not established legal limits for mycotoxin presence in quinoa seeds. However, considering the use of this food as a pseudocereal, and therefore taking into account also for quinoa the maximum levels of mycotoxins established for unprocessed cereals by the EU Commission Regulation EC No 1881/2006 [80], in the work of Ramos-Diaz et al. [23], only one sample (EU origin) exceeded the limit for one mycotoxin (OTA). However, the same authors showed that cleaning methods, such as washing for saponin removal, significantly reduced mycotoxin contamination in quinoa seeds below detection levels. This fact may also explain the total absence of secondary metabolites in the samples analyzed in this research that were processed before commercialization (samples 1–20), as well as the detection of few secondary metabolites in the other samples that were marketed in bulk without receiving, apparently, any type of processing before commercialization (samples 21–25). This supports the idea that processed and marketed quinoa seeds could be generally considered a food matrix with a very low mycotoxin contamination risk, despite the presence of mycotoxigenic fungi. Among the few metabolites identified in samples 21–25, the *Alternaria* metabolite altersetin is reported in the literature for its activity against pathogenic gram-positive bacteria and moderate in vivo efficacy in a murine sepsis model [81,82]. To date, to our knowledge, no toxic effects on humans and animals have been reported, and this is not among the metabolites considered in the dietary assessment exposure to *Alternaria* mycotoxins in Europe [83]. About tryptophol, it is produced by a high number of living organisms, including not only fungi, but also plants, and has several biological activities, being a growth factor in plants; an apoptosis inducer in oomycetes, fungi and human lymphocytes; and is involved in sleeping sickness in humans [84].

The isolation method showed a significant effect on the number of the isolated fungi, with the DFB method allowing obtaining a significantly higher number of colonies of total fungi and of those of the genera *Alternaria*, *Penicillium* and *Fusarium*. Previous studies conducted on seeds of several species reported the higher ability of the DFB method to promote *Fusarium* spp. development [85]. Further, *Alternaria* was previously described as promoted in its development from durum wheat grains by the DFB method rather than PDA [86]. Conversely, the adoption of the PDA method allowed obtaining a significantly higher number of *Aspergillus* colonies. Finally, the average number of *Cladosporium* colonies did not significantly differ between the two isolation methods. The differences between the two methods were not only quantitative, but also qualitative. For example, some species, such as *A. chartarum*, *A. abundans* and *P. verrucosum*, were only obtained with the DFB method, while the species *A. infectoria*, *A. jensenii*, *A. fumigatus* and *P. citreosulfuratum* only on PDA. Therefore, as already described for durum wheat grains [86], considering the effect of the isolation technique in terms of growth promotion of some fungal genera/species rather than others, the combined use of both isolation methods would be desirable also for quinoa seeds to obtain the greatest amount and variability of fungal microorganisms associated with this food matrix.

About the farming system, a higher level of contamination by total fungi and by those of the genera *Cladosporium* and *Penicillium* were observed in the organic farming samples, while those coming from the IPM system showed higher contamination by species of the *Aspergillus* genus. In qualitative terms, the species *A. alternata*, *A. arborescens*, *P. polonicum*, *P. dipodomyis*, *P. chrysogenum*, *A. flavus*, *A. tubingenis*, *C. cladosporiodes*, *C. parasubtilissimum*, *C. uwebraunianum* and *F. oxysporum* were detected from samples obtained both by organic and IPM farming systems, while *A. abundans*, *P. verrucosum* and *P. citreosulfuratum* were detected just in the organic samples, and *A. chartarum*, *A. infectoria*, *A. fumigatus*, *A. jensenii* and *C. allicinum* just in those from IPM system. In total, 14 fungal species were detected in the organic samples and 16 in the samples from IPM ones. In the available literature, discordant data about the level of contamination by mycotoxigenic fungi and mycotoxins in fresh and processed vegetables obtained in organic or non-organic (conventional or IPM) farming systems are present [21,87,88,89]. However, according also to González et al. [87] and Sacco et al. [21], the data obtained in this research support the increased risk of mycotoxigenic fungi in organic vegetable commodities.

In addition, the packaging showed a significant effect on fungal presence, with packaged samples surprisingly showing a higher level of contamination than unpackaged ones marketed in bulk. The higher contamination level in packaged samples is in accordance with those reported in previous studies [21,90]. For sealed plastic packages, this could be explained by the lack of air exchanges with the external environment, which also means: (a) lack of further contamination by external fungi, with the consequence that the level of contamination depends on the quality of the packaged material; (b) creation of humidity content inside the packages, which contributes to the development of mycobioma already present on the material. The lower humidity level in unsealed cardboard boxes could explain the lower contamination level with respect to sealed plastic packages. Further studies about mycotoxin contamination trends following prolonged storage times of processed seeds under different environmental conditions could help to clarify whether post-processing fungal contaminations can alter quinoa seeds.

## 5. Conclusions

The processed and marketed quinoa seeds analyzed in this survey could be considered safe and healthy due to their low mycotoxin contamination risks. This aspect, if added to others, such as the absence of gluten, the low glycemic index and the high nutritional values, increases the importance of this food matrix. Farming systems and packaging may also have a role in the healthiness of this foodstuff.

## Figures and Tables

**Figure 1 pathogens-12-00418-f001:**
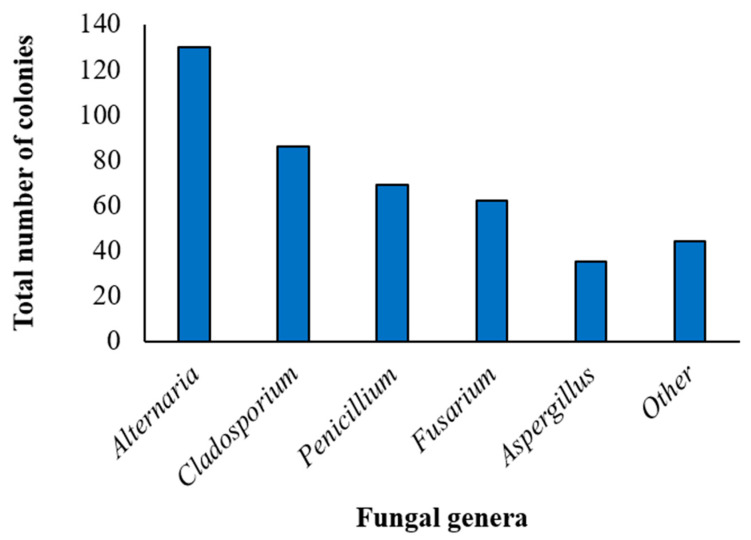
Incidence of different fungal genera in the marketed quinoa seed samples. Incidence of the *Alternaria*, *Cladosporium*, *Penicillium*, *Fusarium*, *Aspergillus* and “other” genera in the total of the 418 colonies obtained by the combination of the 2 isolation methods (isolation on Potato Dextrose Agar, and the deep-freezing blotter method-DFB) from the marketed quinoa seed samples as identified by morphological features (genus).

**Figure 2 pathogens-12-00418-f002:**
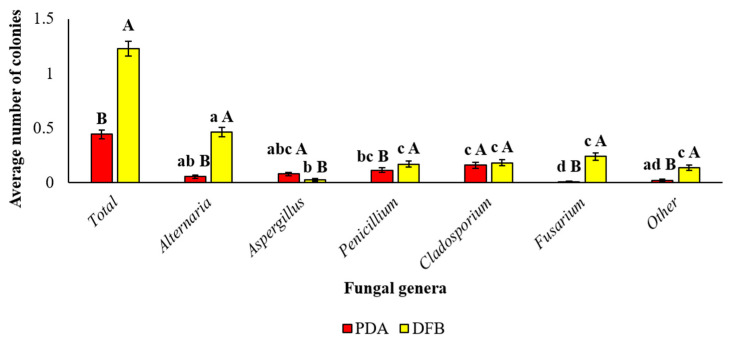
Average number of colonies of each fungal genus isolated from the marketed quinoa seed samples obtained by PDA or DFB methods. Average number of colonies of each fungal genus isolated from the 25 marketed quinoa seed samples as identified by stereomicroscope and microscope observation on colonies of 7 days. Each column represents the average (±SE) of the 25 analyzed samples. Lowercase letters indicate significant differences (*p* < 0.05) between fungal genera within each isolation method (PDA or DFB); uppercase letters indicate significant differences (*p* < 0.05) between the two different isolation methods (PDA or DFB) within each fungal genus.

**Figure 3 pathogens-12-00418-f003:**
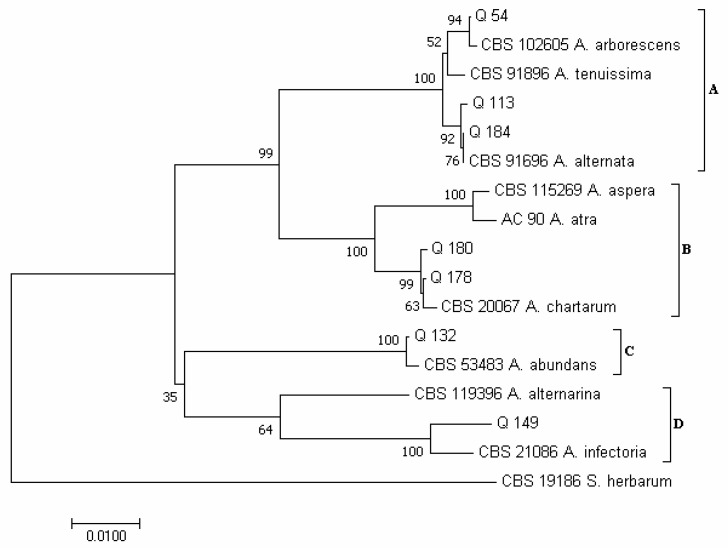
Phylogeny of *Alternaria* isolates obtained from the marketed quinoa seed samples. The evolutionary history of *Alternaria* isolates was inferred using the Neighbor-Joining method [58] and combined dataset of *ITS* and *RPB2* regions sequences. The optimal tree with the sum of branch length = 0.29448430 is shown. The percentage of replicate trees in which the associated *taxa* clustered together in the bootstrap test (1000 replicates) are shown next to the branches [59]. The tree is drawn to scale, with branch lengths in the same units as those of the evolutionary distances used to infer the phylogenetic tree. The evolutionary distances were computed using the Maximum Composite Likelihood method [60] and are in the units of the number of base substitutions per site. The analysis involved 17 nucleotide sequences. All positions containing gaps and missing data were eliminated. There was a total of 878 positions in the final dataset. Evolutionary analyses were conducted in MEGA7 [46].

**Figure 4 pathogens-12-00418-f004:**
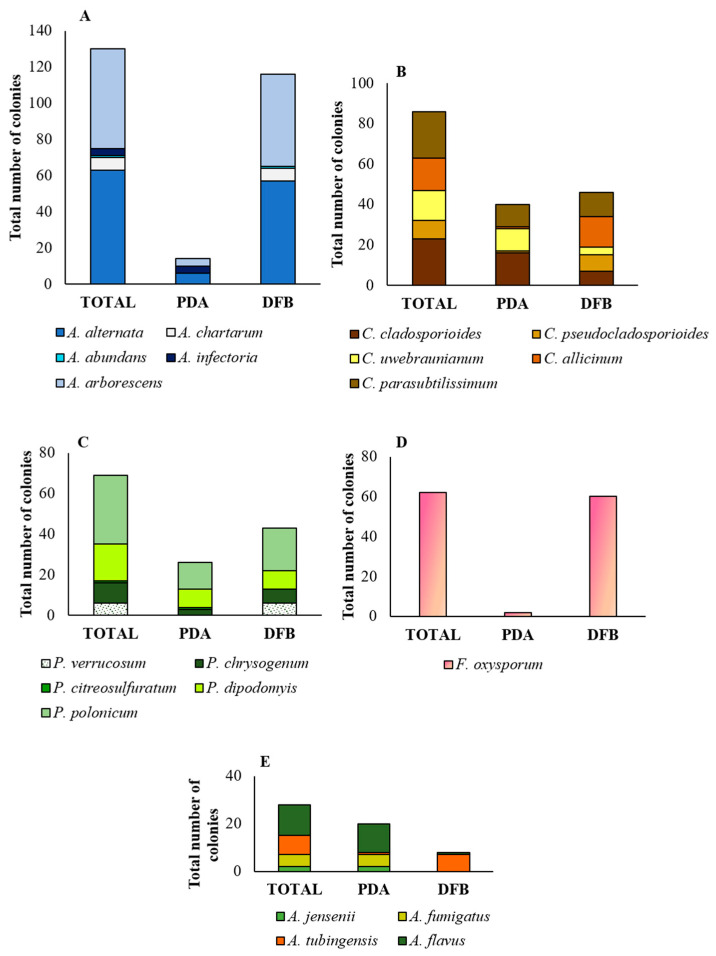
Incidence of fungal genera and species in the marketed quinoa seed samples. Incidence (as numbers of colonies) of the *Alternaria* (**A**), *Cladosporium* (**B**), *Penicillium* (**C**), *Fusarium* (**D**) and *Aspergillus* (**E**) genera and of the relative species obtained from the marketed quinoa seed samples as identified by morphological features (genus) and phylogenetic analysis (species). PDA = colonies obtained by the Potato Dextrose Agar (PDA) method; DFB = colonies obtained by the deep-freezing blotter test; TOTAL = sum of the colonies obtained by isolation on PDA and DFB.

**Figure 5 pathogens-12-00418-f005:**
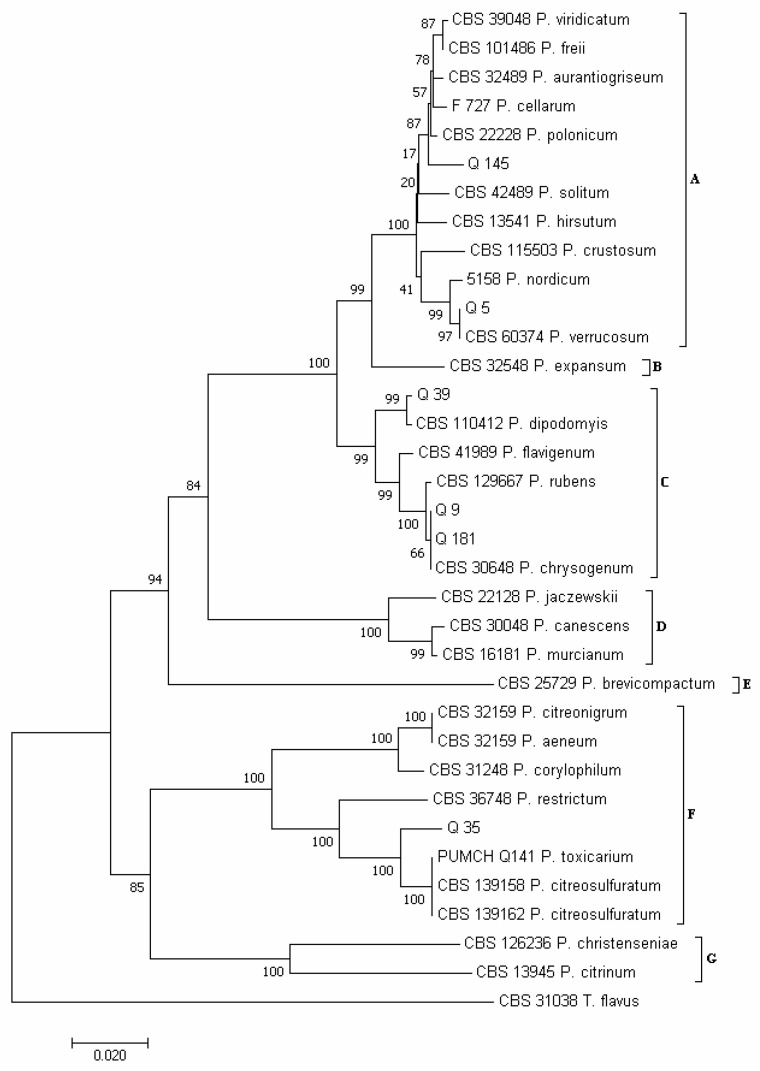
Phylogeny of *Penicillium* isolates obtained from the marketed quinoa seed samples. The evolutionary history of *Penicillium* isolates was inferred using the Neighbor-Joining method [58] and combined dataset of partial *ITS* and *RPB2* regions sequences. The optimal tree with the sum of branch length = 0.81966305 is shown. The percentage of replicate trees in which the associated *taxa* clustered together in the bootstrap test (1000 replicates) are shown next to the branches [59]. The tree is drawn to scale, with branch lengths in the same units as those of the evolutionary distances used to infer the phylogenetic tree. The evolutionary distances were computed using the Maximum Composite Likelihood method [60] and are in the units of the number of base substitutions per site. The analysis involved 35 nucleotide sequences. All positions containing gaps and missing data were eliminated. There was a total of 854 positions in the final dataset. Evolutionary analyses were conducted in MEGA7 [46].

**Figure 6 pathogens-12-00418-f006:**
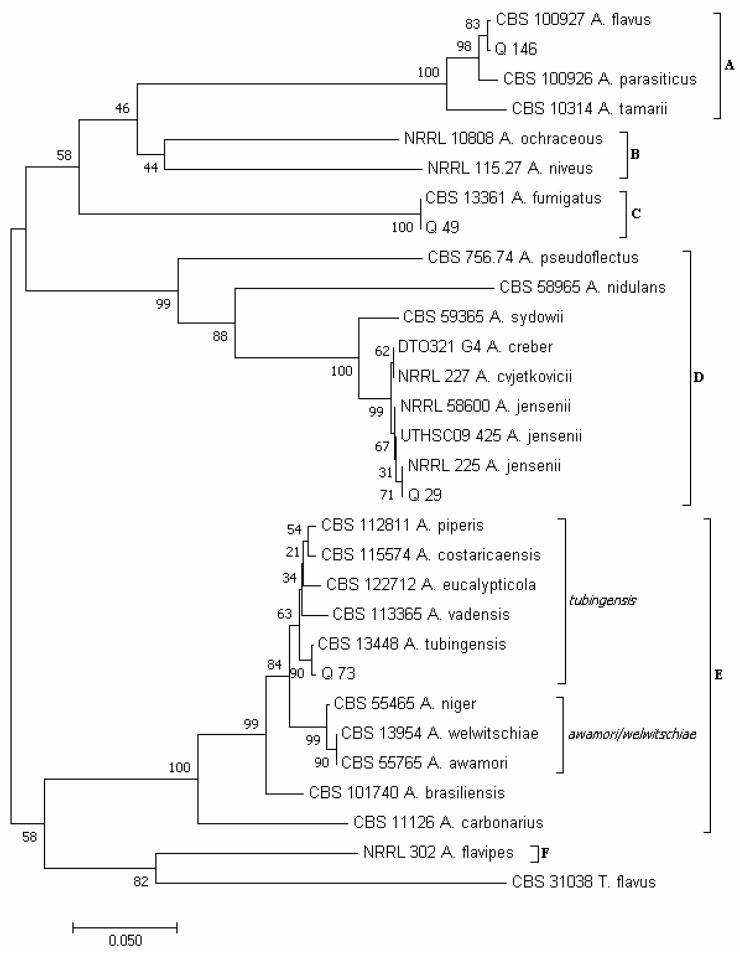
Phylogeny of *Aspergillus* isolates obtained from the marketed quinoa seed samples. The evolutionary history of *Aspergillus* isolates was inferred using the Neighbor-Joining method [58] and combined dataset of *BenA* and *CaM* gene sequences. The optimal tree with the sum of branch length = 1.72566190 is shown. The percentage of replicate trees in which the associated *taxa* clustered together in the bootstrap test (1000 replicates) are shown next to the branches [59]. The tree is drawn to scale, with branch lengths in the same units as those of the evolutionary distances used to infer the phylogenetic tree. The evolutionary distances were computed using the Maximum Composite Likelihood method [60] and are in the units of the number of base substitutions per site. The analysis involved 30 nucleotide sequences. All positions containing gaps and missing data were eliminated. There was a total of 398 positions in the final dataset. Evolutionary analyses were conducted in MEGA7 [46].

**Figure 7 pathogens-12-00418-f007:**
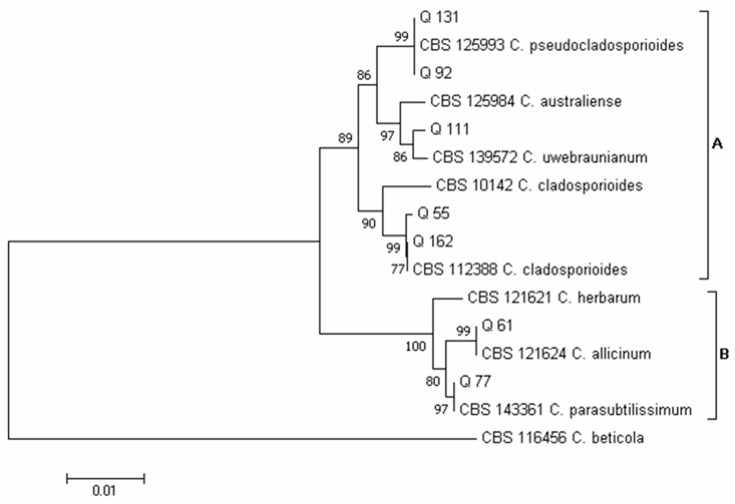
Phylogeny of *Cladosporium* isolates obtained from the marketed quinoa seed samples. The evolutionary history of *Cladosporium* isolates was inferred using the Neighbor-Joining method [58] and combined dataset of partial *ITS* and *ACT* gene sequences. The optimal tree with the sum of branch length = 0.16365539 is shown. The percentage of replicate trees in which the associated *taxa* clustered together in the bootstrap test (1000 replicates) are shown next to the branches [59]. The tree is drawn to scale, with branch lengths in the same units as those of the evolutionary distances used to infer the phylogenetic tree. The evolutionary distances were computed using the Maximum Composite Likelihood method [60] and are in the units of the number of base substitutions per site. The analysis involved 16 nucleotide sequences. All positions containing gaps and missing data were eliminated. There was a total of 548 positions in the final dataset. Evolutionary analyses were conducted in MEGA7 [46].

**Figure 8 pathogens-12-00418-f008:**
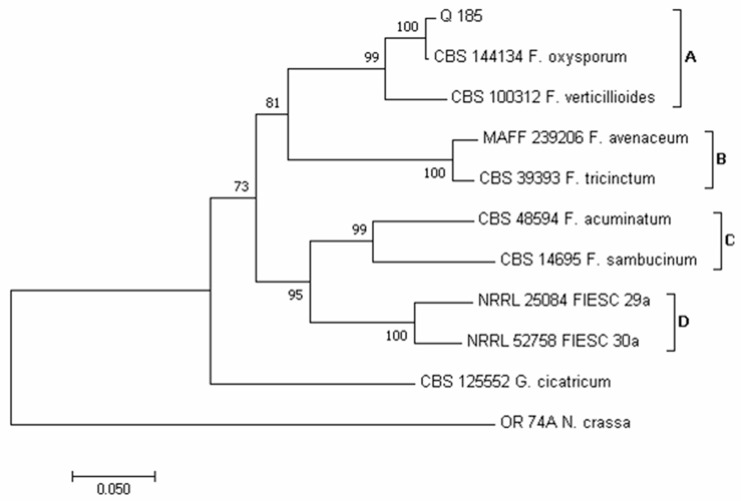
Phylogeny of *Fusarium* isolates obtained from the marked-bought quinoa seed samples. The evolutionary history of *Fusarium* isolate was inferred using the Neighbor-Joining method [58] and dataset of *TEF1-α* gene sequences. The optimal tree with the sum of branch length = 1.17466622 is shown. The percentage of replicate trees in which the associated *taxa* clustered together in the bootstrap test (1000 replicates) are shown next to the branches [59]. The tree is drawn to scale, with branch lengths in the same units as those of the evolutionary distances used to infer the phylogenetic tree. The evolutionary distances were computed using the Maximum Composite Likelihood method [60] and are in the units of the number of base substitutions per site. The analysis involved 11 nucleotide sequences. All positions containing gaps and missing data were eliminated. There was a total of 506 positions in the final dataset. Evolutionary analyses were conducted in MEGA7 [46].

**Figure 9 pathogens-12-00418-f009:**
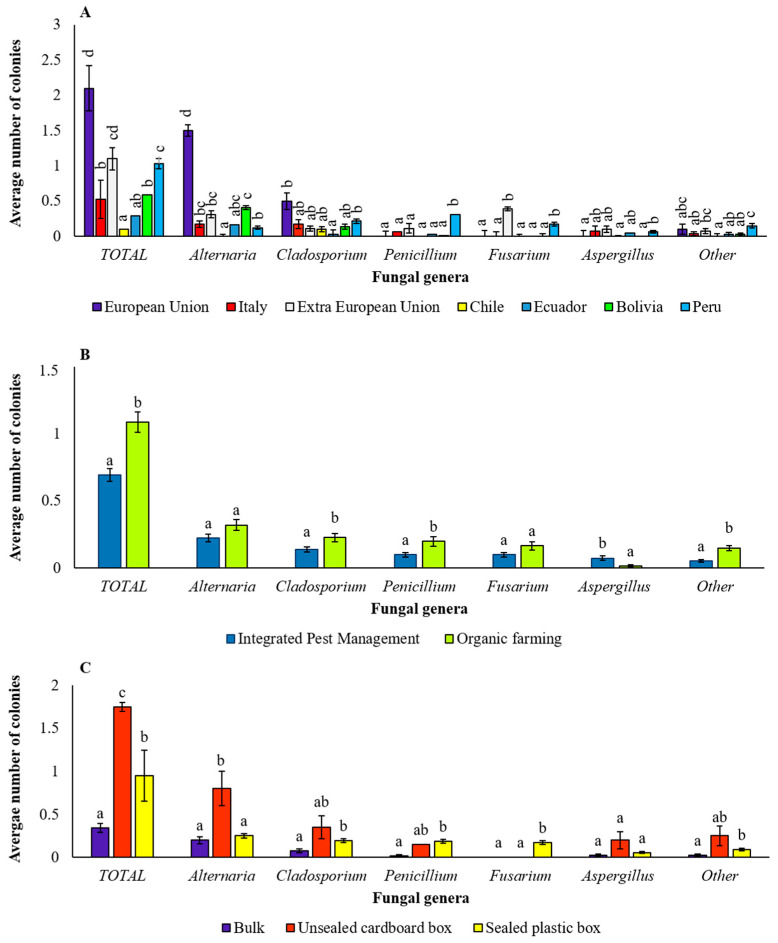
Incidence of total isolated fungi and of fungi of each genus depending on origin, farming system and packaging. Average number of colonies of total isolated fungi and of each fungal genus in samples with different origin (**A**), farming system (**B**) and packaging (**C**). Bars represent the average (±SE); different letters indicate significant differences (*p* < 0.05) within total or each genus.

**Table 1 pathogens-12-00418-t001:** Marketed quinoa seed samples analyzed in this study.

Sample	Origin	Seed Color	Farming	Packaging
1	Extra EU (Peru)	White	Integrated	Sealed Plastic Box
2	EU (Italy)	White	Integrated	Sealed Plastic Box
3	EU (Italy)	White	Integrated	Sealed Plastic Box
4	Extra EU (Peru)	White	Organic	Sealed Plastic Box
5	Extra EU (Country not specified)	White	Integrated	Sealed Plastic Box
6	Extra EU (Peru)	White	Organic	Sealed Plastic Box
7	EU (Italy)	White	Integrated	Sealed Plastic Box
8	EU (Italy)	White	Integrated	Sealed Plastic Box
9	Extra EU (Country not specified)	White	Organic	Sealed Plastic Box
10	Extra EU (Country not specified)	White	Integrated	Unsealed Cardboard Box
11	Extra EU (Peru)	White	Integrated	Bulk
12	Extra EU (Country not specified)	White	Organic	Sealed Plastic Box
13	Extra EU (Peru)	White	Integrated	Sealed Plastic Box
14	Extra EU (Peru)	Black	Integrated	Sealed Plastic Box
15	Extra EU (Peru)	Red	Integrated	Sealed Plastic Box
16	Extra EU (Peru)	White	Integrated	Sealed Plastic Box
17	EU (Country not specified)	White	Organic	Sealed Plastic Box
18	Extra EU (Bolivia)	Black	Organic	Sealed Plastic Box
19	Extra EU (Bolivia)	White	Organic	Sealed Plastic Box
20	Extra EU (Peru)	White	Organic	Sealed Plastic Box
21	Extra EU (Bolivia)	White	Organic	Bulk
22	Extra EU (Chile)	White	Integrated	Bulk
23	Extra EU (Ecuador)	White	Integrated	Bulk
24	Extra EU (Ecuador)	White	Integrated	Bulk
25	Extra EU (Bolivia)	White	Integrated	Bulk

**Table 2 pathogens-12-00418-t002:** Fungal genera isolated from marketed quinoa seed samples.

	Sample																								
	1	2	3	4	5	6	7	8	9	10	11	12	13	14	15	16	17	18	19	20	21	22	23	24	25
*Alternaria*	● *	●	●			●	●	●		●		●	●	●	●	●	●	●	●	●			●	●	●
*Cladosporium*	●	●	●	●	●	●	●	●		●	●	●	●	●	●		●	●	●	●	●	●	●		●
*Penicillium*	●	●		●		●		●	●	●			●	●	●	●				●	●		●		
*Aspergillus*					●	●	●	●		●	●				●	●				●			●	●	
*Fusarium*					●	●									●					●					

* Black dots indicate the samples where each genus were detected.

**Table 3 pathogens-12-00418-t003:** Secondary metabolites detected in the frieze-dried culture of the fungal isolates Q 145 of *Penicillium polonicum* obtained from marketed quinoa (*Chenopodium quinoa* Willd.) seed samples and grown on Czapek Yeast Autolysate (CYA) Agar medium.

*Penicillium* Secondary Metabolites	μg Kg^−1^
**Anthraquinoids**	
Endocrocin	158
**Asperphenamates**	
Asperphenamate	377
**Diketopiperazines**	
Brevianamide F	5510
*Cyclo* (L-Pro_L-Tyr)	32,900
*Cyclo* (L-Pro_L-Val)	20,400
Neoxaline	1350
Roquefortine C	4870
Roquefortine D	146
Puberulin A	46,400
Rugulosuvin	17,900
**Imidazopyridoindoles**	
Meleagrin	1670
**Penicillic acids**	
Penicillic acid	404,000
**Propionic acids**	
3-Nitropropionic acid	1,300,000
**Terpenes**	
Andrastin A	29,000
Andrastin B	1850
Andrastin C	45,900
Mycophenolic acid	4780
**Viridicatols**	
Cyclopenin	106,000
Cyclopenol	335,000
Cyclopeptine	27,500
Dehydrocyclopeptine	27,100
O-Methylviridicatin	36,800
Viridicatin	17,800
Viridicatol	428,000

**Table 4 pathogens-12-00418-t004:** Secondary metabolites (μg kg^−1^) detected in five (21–25) marketed quinoa (*Chenopodium quinoa* Willd.) seed samples. No secondary metabolites were detected in the other twenty (1–20) samples, so these were not included in the table.

Secondary Metabolites	Seed Samples				
	21	22	23	24	25
**Tetramic acid derivatives**					
Altersetin	30.6	1.18	13.1	4.96	8.56
**Diketopiperazines**					
Brevianamide F	3.53	0.25	1.92	3.55	2.25
*Cyclo* (L-Pro L-Tyr)	12.6	2.18	15.3	12.3	10.2
*Cyclo* (L-Pro_L-Val)	12.7	3.24	21.6	13.2	11.4
Rugulosuvin	1.04	<LOD	<LOD	1.06	<LOD
**Indole alcohol**					
Tryptophol	49.2	5.79	10.4	97.5	24.7

## Data Availability

All data supporting the conclusions of this research manuscript are included in this manuscript and its additional files.

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
