# Peer review of "Marketed Quinoa (Chenopodium quinoa Willd.) Seeds: A Mycotoxin-Free Matrix Contaminated by Mycotoxigenic Fungi"

_pathogens, 2023, doi:10.3390/pathogens12030418_

Round 1

Reviewer 1 Report

Dear author

This study showed the contamination of marketed quinoa with fungi and mycotoxins and compares the effects of geographical origin, farming system and packaging forms on fungal contamination. The manuscript fits within the scope of the journal. The study methods are explained clearly. The conclusions or summary are supported by the content. The author's work on discussing achieved results is appreciated.

I have some major recommendations for authors:

(1)Keywords suggested to delete the specific strain name, add quinoa”.

Introduction 

(2) Line 34-88 Too much information about quinoa's origins, evolutionary history, nutrition, etc., should be condensed into one paragraph.

(3)There is a formatting error on line 103.

Materials and Methods

(4)Are samples taken immediately for testing? If not, please add storage conditions after sample collection.

Results

(5)How is the total calculated in Figure 1, please indicate in the note.

(6)In line 271, in the statistical analysis, P < 0.05 means a significant difference, not P = 0.05. Please correct the full text.

(7)In Figure 4, the total number of colonies found should not be the sum of the number of colonies found in way PDA and way DFB, because the species found in way PDA and way DFB have the same part. Total = PDA + DF - PDA-DFB common. If the value cannot be calculated, delete the representation of the total.

Discussion

(8)This study found that there was fungal contamination in quinoa, but no mycotoxins. This phenomenon is hard to understand. The discussion should focus on the possible reasons for this phenomenon. In addition. Mycotoxins can also be produced if fungal contamination is present, provided conditions are suitable for storage. Whether the author has done similar research and could you make some suggestions on the storage and packaging of quinoa to further reduce the risk of mycotoxin contamination?

Author Response

Dear reviewer, 

please find attachment the response. 

Thanks. 

Reviewer 2 Report

This manuscript presents the results of a study on the mycobiome associated with marketed quinoa seeds and its mycotoxin contamination potential. Considering the increasing trend in the use of this kind of food and the lack of regulative parameters concerning its qualitative standards as related to safety for consumers, investigations on the subject are relevant in view of increasing the available information on this topic. However, as a consequence of the minimum levels of mycotoxin contamination which could be detected, contents are basically focused on the identification of the fungal species assortment occurring on the seed surface. In this respect, I remark that there are several aspects requiring substantial revision before it can be accepted for publication.

1)      Authors grouped their isolates in a small number of morphotypes, claiming that they may correspond to definite species. This inference is incorrect, since many related fungal species are morphologically indistinguishable; from which it follows that it is arbitrary to ascribe an isolate to a species just because it is morphologically similar to another one which has been identified through molecular markers.

2)      Approximation in species identification particularly emerges in the case of Penicillium isolates. In fact, authors are unable to classify isolate Q145, and try to appeal to chemotaxonomic inferences. However, they should be cautious in this respect, considering that recent evidence emphasizes the uneven biosynthetic abilities within fungal species, due to defective gene clusters and/or gene silencing. Whatever, the limitation of their method is underlined by the fact that P. aurantiogriseum, P. viridicatum and P. polonicum are not the same species, as it could be deduced from their partial phylogenetic analysis. Indeed, there are more species which are closely related to the above three (e.g. P. christenseniae), and for a circumstantial identification the whole set of related species should be considered.

3)      With reference to the analysis of Penicillium, the inclusion of Talaromyces minioluteus is not acceptable. In fact, since about 10 years the biverticillate Penicillium species have been separated and officially classified in the genus Talaromyces. As the biverticillate condition can clearly result through microscope observation, isolate Q25 should have been included in the group of 'other' fungi, rather than being considered as a 'Penicillium morphotype'.

4)      As a general remark, identification based on phylogenetic analysis is risky when the selection of reference strains is partial and biased. In fact, the biomolecular approach is recently disclosing the existence of an increasing number of sister species within the existing taxa, especially in the case of the fungal genera which were found to be dominant in the present study. I believe that authors should rather rely on the results of blasting their sequences in GenBank. As an example, blasts of sequences of isolate Q146 unequivocally show it belongs to A. flavus, which makes the subsequent metabolomic analysis unnecessary. Hence, I suggest that authors prepare a table considering all the reference isolates, reporting results of GenBank blasting and indicating the most closely related strains/species. Phylograms could be retained as supplementary material. With reference to the captions of phylograms, I observe that, although phylogenetic studies are fundamental in the assessment of the evolutionary history of organisms, it cannot be said that a phylogenetic study represents the 'evolutionary history of isolates'. Actually, these phylograms are merely indicative of the phylogenetic relationships among the isolates recovered from quinoa seeds and some arbitrarily chosen reference strains, and they can only be used in view of an approximate species identification.

Besides criteria followed in species identification, some other aspects require to be revised or reconsidered. With reference to the alleged correlation with the farming system, I remark that it does not make sense to look for a statistical correlation when the collected data are not homogeneous. Indeed, samples referred to the same 'farming system' were collected in very diverse conditions in terms of environment and crop management, which undoubtedly influence fungal occurrence more than the hypothetically uniform farming; making it meaningless to try to find any significant correlation. Moreover, authors grouped some spare fungal genera in the group of ‘other fungi’, which is obviously heterogeneous in its composition and possible impact: what is the relevance of analyzing the occurrence of 'other fungi' in statistical terms?

Concerning mycotoxin contamination, it raises perplexity the finding that fungal metabolites could be detected in 5 samples only, and all of them were contaminated with a small group of products. How do authors comment on this finding? Was any fungal species dominant in all these 5 samples which could be considered responsible for this similar contamination pattern?

Finally, the introduction is too long and wordy; it should be rewritten in a concise way, as this article is not meant to present history and virtues of quinoa.

Although generally correct, the English style also requires an accurate revision. Below a partial list of corrections that hopefully can be useful to authors in view of improving quality of the text.

line 26: you mean 'highlighting that'?

line 44: replace 'including' with 'such as';

line 46: 'crops' and delete 'by several research projects dealing with the adaptation of this species to the peninsula': this phrase is unnecessary specification and incorrect in setting: a plant does not adapt to a 'peninsula', rather to its climatic conditions.

Line 52: I would say 'variable photoperiod' rather than 'several day lengths', and semicolon after 'optimal';

line 53: it is 'mm'!

line 57: considering content, I would rather use 'low' and 'high' instead of 'few' and 'many';

line 59: 'spread' instead of 'distribution';

line 62: 'Year'; do not use italics;

line 71: correct to '...gluten-free, hence suitable for';

line 84: 'pseudocereals' without quotation marks;

line 93: full stop or semicolons after 'humans', followed by 'they are...';

line 100: correct to '...[21]; moreover, multiple...';

lines 103-105: a better form could be 'However, being just an ingredient in these matrices, the impact of quinoa on contamination is uncertain.'

section 2.2: a better description of the isolation procedure based on the DFB method is necessary: was PDA used in this case too?

lines 165-178: this information should be better resumed in a table;

line 263: correct to '...of the two latter genera...';

line 305: 26 isolates? Previous text refers to 25 representative isolates;

line 309: 'four isolates'.

Author Response

Dear reviewer, 

please find in the attachment the response to your comments. 

Thanks. 

Reviewer 3 Report

The present work points out the relevance of constantly monitoring marketed quinoa (Chenopodium quinoa). Specifically, 25 marketed quinoa seeds from several world regions were examined. As the main results, 20 fungal species were isolated, being the first report worldwide in quinoa seeds for some species belonging to fungal genera such as Aspergillus, Fusarium, and Penicillium

The manuscript is well-written, and the structure is in accordance with the issue. The main strength of the manuscript is the well-developed introduction section. Furthermore, the M&M section is very detailed. I consider the survey adequate because 25 samples from several countries, farming systems, types of quinoa, and grain colors were recollected.  Statistical design and analysis are appropriate. Phylogenetic trees for each genus are an excellent complement to this survey. In general, tables and figures are ok. Regarding the results sections, the co-occurrence described by the authors (mainly Alternaria-Cladosporium) is interesting, adding that fungal species contaminated 100% of samples. In concordance, several works emphasized the presence of asymptomatic grains in other crops (such as wheat, barley, triticale, oat, maize, etc.).

Moreover, it is interesting that grains marketed in sealed/unsealed cardboard boxes have higher fungal contamination than grains marketed in bulk. The discussion section is solid, strong, and well-developed too. The conclusion, which supports that the marketed quinoa analyzed is free from mycotoxins, is another novel result. I agree with the authors that the washing for saponins removal could be the cause (sometimes farmers apply up to 7-8 washes).

I only have a few minor suggestions:

- Line 50: Please, unify thousands separator along the complete manuscript. In previous sentences, authors used a dot as separator.

- Some figures, such as Fig. 1 and Fig. 2, could be improved (the chosen visualization is not the best, in my opinion). If the authors consider it necessary, they can modify these figures towards a friendly visualization, similar to Fig. 4  or Fig. 9.

The authors have proven experience in the study area (phytopathology and food safety), being the quality of their works recognized worldwide.  This fact is denoted in the high quality of the present manuscript regarding writing and statistical analysis, which gives it enough solidity to be published. Therefore, I consider that the current version of the manuscript is suitable to be accepted for publication in Pathogens.

Author Response

(The authors gave the same response as above.)

Round 2

Reviewer 1 Report

Dear author

      This article has been revised according to the suggestions, and its quality and readability have been improved. I suggest accepting this article.

Author Response

Dear reviewer, 

thanks for your valuable revision. 

Reviewer 2 Report

Authors have partially accepted my comments and suggested revisions, which means that I cannot be completely satisfied with the quality of this version of the manuscript. Indeed, the issue of taxonomic identification is very delicate, as it is demonstrated by the introduced amendments. In this respect authors should again update identification of strain Q35 as P. toxicarium, since this species is invalid and not included in the revised list of Penicillium species (cfr. www.sciencedirect.com/science/article/pii/S0166061620300129); rather, it is regarded as a synonym of P. citreosulfuratum (cfr. link.springer.com/article/10.5598/imafungus.2016.07.01.06) or P. spinulosum (cfr. Mycobank).

Author Response

Dear reviewer, 

please find in the attached file our response to your comments.

Thanks. 
